# Temporal Dynamics of an Asymmetrical Dielectric Nanodimer Wrapped with Graphene

Xinchen Jiang [1], Yang Huang [2], Pujuan Ma [3,*], Alexander S. Shalin [4,5,6] and Lei Gao [1,4,*]

1 School of Physical Science and Technology & Collaborative Innovation Center of Suzhou Nano Science and Technology & Jiangsu Key Laboratory of Thin Films, Soochow University, Suzhou 215006, China
2 School of Science, Jiangnan University, Wuxi 214122, China
3 School of Physics and Electronics, Shandong Normal University, Jinan 250014, China
4 School of Optical and Electronic Information, Suzhou City University, Suzhou 215104, China
5 Center for Photonics and 2D Materials, Moscow Institute of Physics and Technology, Dolgoprudny 141700, Russia
6 Faculty of Physics, M. V. Lomonosov Moscow State University, Moscow 119991, Russia
* Correspondence: pujuan-ma@sdnu.edu.cn (P.M.); leigao@suda.edu.cn (L.G.)

**Abstract:** We theoretically and numerically investigate the temporal dynamics of a nanodimer system consisting of a pair of graphene-wrapped dielectric nanospheres with tunable radii. Considering that symmetry breaks on resonant frequencies, we derive the temporal kinetic equations in an asymmetric form by utilizing the dispersion relation method in dipole limit. The bifurcation diagrams achieved via the analysis on the linear instability and numerical solutions can quantitatively characterize the complex coexistences of stationary and dynamical behaviors in this dimer system, and the asymmetry apparently can increase the number of regimes with the periodic self-oscillation state or chaos. Furthermore, we find that the indefinite switching not only can be triggered among the stationary steady solutions, but it also universally exists among all the possible solutions in a coexistent regime. The switching can be tuned by applying a hard excitation signal with different durations and saturation values. Our results may provide new paths to realize a nonlinear nanophotonic device with tunable dynamical responses or even multi-functionalities.

**Keywords:** nanodimer; temporal dynamics; asymmetry; dipole–dipole limit; graphene; nonlinearity

## 1. Introduction

Nowadays, it is well known that nonlinearity is widely involved in a variety of microelectronics devices, such as logic gates, frequency filter, data storage, signal generators, and even more complicated Chua's circuit [1,2]. In view of all-optical information processing, the concept of lumped optical nanoelements has become a field of great concern in the regime of on-chip photonic circuitry [3–11], so optical nonlinearity may similarly be expected to provide various functionalities as an extension of metactronics paradigm, whose operating frequency can range from GHz to infrared (IR) and visible regime. In recent years, several pioneer works have focused on this topic. Based on plasmonic two-wire transmission-line (TWTL) architecture, half-subtractor and demultiplexer have been realized experimentally [12]. And the X-shaped plasmonic microstructures with cover layer can provide low-power half- and full-adders features with small footprint [13]. By utilizing two counter propagating frequency combs with temporally synchronized pulses, the excitation of multiple electrically tunable plasmons in fiber can demonstrate the potential for integrated logic operations [14]. For nanoparticle systems, the terahertz radiation generation based on modulation instability can be a pioneer work [15], and the mechanism can lead to the periodic rotation and switching of the scattering pattern acting as a nanoantenna [16]. By utilizing the magneto-optic material, tunable Fano switching sensitive to light circular polarization has been proposed [17]. Recently, it was found that a nonlinear

nanodimer made of a pair of resonant nanoparticles can operate as tristable and astable multivibrators as well as random generators depending on the external stimulation [18,19]. Furthermore, the chains or arrays of nonlinear nanoparticles have also been extended to the propagation of a wave-packet, leading to interesting nonlinear phenomena, such as plasmon oscillons [20], solitons [21–23], Faraday waves [24] and chaos [25].

Notably, as a simple but universal model, the coupled dimer system has attracted more and more attention due to the ability of operation on localized surface plasmon resonances (LSPRs) [26,27]. Such a system has been potentially applied to enhance the Raman spectroscopy, optical tweezers [28–30], optical switches [31] and bio-sensing [32]. As symmetry plays an important role in our understanding of the physics in plasmonic dimers, abundant and novel physical phenomena can be anticipated when the symmetry of a dimer is broken, including directional photon-sorting [33], Fano resonances [34], indefinite switching [35], temporal beats with long-lasting tail [36], neuronlike spiking dynamics [37] and more [38,39]. Hence, the introduction of asymmetry may provide new degrees of freedom for modulation on nonlinear temporal dynamics.

In this work, we propose a nanodimer system consisting of a pair of graphene-wrapped dielectric nanosphere with Kerr-type nonlinearity, and consider the influence of asymmetry on the optical response by varying the radius of one sphere. By utilizing the dispersion relation method [15,17,19,20], we derive the temporal kinetic equations in an asymmetric form. Further, according to the linear instability analysis and numerical calculation, we present the bifurcation diagrams and characterize the complex dynamical behaviors by means of quantitative mathematical tools of nonlinear dynamics theory [40,41]. Eventually, we explore the interesting indefinite switching [35] among the temporal dynamical and stationary steady solutions by introducing a hard excitation signal due to the coexistence of these solutions.

## 2. Materials and Methods

As shown in Figure 1, the asymmetric nanodimer dynamical system includes two nonlinear graphene-wrapped subwavelength nanospheres with radii $r_{1,2}$, center-to-center distance $d$ and submerged in host medium, under the illumination of an external optical field $E_0$ with a frequency $\omega$, close to the frequencies of the surface plasmon resonances of nanospheres. The relative permittivities of two dielectric nanospheres and the host medium are denoted as $\varepsilon_{1,2}$ and $\varepsilon_h$ respectively. Since the graphene can be treated as a monolayer, we consider the graphene as a 2D material with conductivity $\sigma = \sigma_L + \sigma_{NL}|E_c|^2$, where $\sigma_L$ and $\sigma_{NL}$ are the linear and Kerr-nonlinear parts of the conductivity, respectively, and $|E_c|^2$ is the local field intensity of the interface between the individual nanosphere and the host medium. Generally, in the terahertz range, the graphene can be well described in a Drude-like form at room temperature, which leads to the simplified $\sigma_L$ and $\sigma_{NL}$ written as follows [42,43]:

$$\sigma_L(\omega) = \frac{ie^2 E_F}{\pi\hbar^2(\omega + i\xi^{-1})}, \quad \sigma_{NL}(\omega) = \frac{-i9e^4 v_F^2}{8\pi E_F \hbar^2 \omega^3} \tag{1}$$

where $e$, $\hbar$, $\xi$, $E_F$ and $v_F$ are electron charge, reduced Planck constant, electron–phonon relaxation time, Fermi energy and Fermi velocity, respectively.

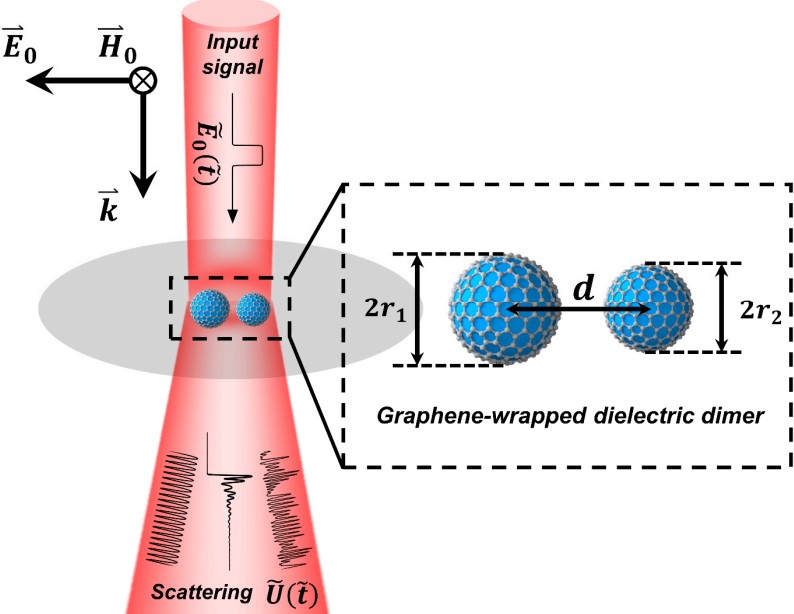

**Figure 1.** Schematic diagram of the graphene-wrapped dielectric dimer with radii $r_{1,2}$. The dimensionless external field $\widetilde{E}_0$ for stimulation can be either a background field $\widetilde{E}_b$ or a combination of $\widetilde{E}_b$ and a hard excitation $\widetilde{E}_{pulse}(\widetilde{t})$. The corresponding dimensionless forward scattering intensity $\widetilde{U}(\widetilde{t})$ has also been presented here for the possible stationary or temporal outputs.

In the condition of $r_{1,2}/d \leq 1/3$, one can start the coupled expressions in the following form by employing the point dipole approximation:

$$p_{1,2} = \alpha_{1,2}(\omega)(E_0 + Gp_{2,1}) \tag{2}$$

where $E_0$ is the external field including the background field $E_b$ and hard excitation $E_{pulse}$ and $\alpha_{1,2}(\omega)$ are the electric polarizabilities of the individual graphene-wrapped nanospheres and can be derived according to Ref. [44] in consideration of the radiation effect:

$$\alpha_{1,2}(\omega) = 4\pi\varepsilon_0\varepsilon_h\left\{\frac{\varepsilon_{1,2} + 2\varepsilon_h + 2\Theta_{1,2}(\omega)}{r_{1,2}^3[\varepsilon_{1,2} - \varepsilon_h + 2\Theta_{1,2}(\omega)]} - i\frac{2}{3}k^3\right\}^{-1} \tag{3}$$

with $\Theta_{1,2}(\omega) = i\left(\sigma_L + \sigma_{NL}|E_c|^2\right)/(\omega r_{1,2}\varepsilon_0)$, $\varepsilon_0$ is the vacuum permittivity, $k$ is the wave number, and $G$ describes the dipole–dipole coupling between the nanospheres, depending on the direction of the wave vector $\vec{k}$. In this work, we consider $\vec{k}$ as perpendicular to the dimer axis, which results in $G = e^{ikd}(ikd - 1)/(\varepsilon_h d^3)$ [16]. So this is a relatively strong coupling case compared with the one in the case of $\vec{k}$ being parallel to the dimer axis, and may bring more complicated nonlinear behaviors. In the linear situation without coupling, one can expect the dipole resonant frequency of an individual nanosphere by setting the denominator of $\alpha_{1,2}(\omega)$ to zero, and the result would be as follows:

$$\omega_{1,2} = \left[\frac{2e^2 E_F}{r_{1,2}\varepsilon_0\pi\hbar^2(\varepsilon_{1,2} + 2\varepsilon_h)} - \zeta^{-2}\right]^{1/2} \tag{4}$$

which is only radius-dependent if we consider the parameters of graphene and the dielectric constant $\varepsilon_{1,2}$ of the cores to be the same.

Following the spirit of the dispersion relation method [15,17,19,20], we further derive the dynamical system in the assumption of weak nonlinearity, dissipation and detuning,

i.e., $\sigma_{NL}|E_c|^2 \ll \sigma_L$, $\mathrm{Im}(\sigma_L) \ll \mathrm{Re}(\sigma_L)$ and $(\omega - \omega_{1,2})/\omega_{1,2} \ll 1$, respectively. One can decompose $\alpha_{1,2}^{-1}(\omega)$ in the vicinity of $\omega_{1,2}$, respectively, and keep the first-order terms involving time derivatives [23,25,45],

$$\alpha_{1,2}^{-1}(\omega) \approx \alpha_{1,2}^{-1}(\omega_{1,2}) + \left.\frac{\mathrm{d}\alpha_{1,2}^{-1}(\omega)}{\mathrm{d}\omega}\right|_{\omega=\omega_{1,2}} \left(\Delta\omega_{1,2} + i\frac{\mathrm{d}}{\mathrm{d}t}\right) \tag{5}$$

where $\Delta\omega_{1,2} = \omega - \omega_{1,2}$ are the detuning variables.

By substituting Equation (5) into Equation (2) and applying dimensionless processing, the nonlinear coupled dimensionless kinetic equations can be written as follows:

$$\begin{cases} W_1 \cdot i\frac{d\widetilde{P}_1}{d\widetilde{t}} + \left(i\widetilde{\gamma}_1 + W_1\widetilde{\Omega}_1 + |\widetilde{P}_1|^2\right)\widetilde{P}_1 + \widetilde{G}\widetilde{P}_2 = \widetilde{E}_0 \\ W_2\frac{\omega_1}{\omega_2} \cdot \varsigma i\frac{d\widetilde{P}_2}{d\widetilde{t}} + \left(i\widetilde{\gamma}_2 + W_2\widetilde{\Omega}_2 + \varsigma^2|\widetilde{P}_2|^2\right)\varsigma\widetilde{P}_2 + \kappa\widetilde{G}\widetilde{P}_1 = \kappa\widetilde{E}_0 \end{cases} \tag{6}$$

where $W_{1,2} = \omega_{1,2}^2/\left(\omega_{1,2}^2 + \xi^{-2}\right)$, $\widetilde{\Omega}_{1,2} = (\omega - \omega_{1,2})/\omega_{1,2}$ and $\widetilde{t} = \omega_1 t$ indicate the dimensionless constants, detuning parameters and time unit, respectively. The dipole orientation of $\widetilde{P}_{1,2}$ is along the dimer axis. Furthermore, the other dimensionless parameters and variables in Equation (6) can be expressed as,

$$\widetilde{P}_{1,2} = \frac{p_{1,2}}{[2(\varepsilon_1 + 2\varepsilon_h)]^{1/2}\varepsilon_h r_1^3}\sqrt{\frac{2i\sigma_{NL}(\omega_1)}{\omega_1 r_1 \varepsilon_0}} \tag{7a}$$

$$\widetilde{E}_0 = -3\varepsilon_h\sqrt{\frac{2i\sigma_{NL}(\omega_1)}{\omega_1 r_1 \varepsilon_0}} \cdot \frac{E_0}{[2(\varepsilon_1 + 2\varepsilon_h)]^{3/2}} \tag{7b}$$

$$\widetilde{\gamma}_{1,2} = \frac{\xi^{-1}}{2\omega_{1,2}} + \frac{\varepsilon_h(kr_{1,2})^3}{\varepsilon_{1,2} + 2\varepsilon_h}, \quad \widetilde{G} = \frac{3\varepsilon_h}{(\varepsilon_1 + 2\varepsilon_h)}\left(\frac{r_1}{d}\right)^3(1 - ikd)e^{ikd} \tag{7c}$$

$$\varsigma = \left(\frac{r_1}{r_2}\right)^{3.5}\sqrt{\frac{\omega_1\sigma_{NL}(\omega_2)}{\omega_2\sigma_{NL}(\omega_1)}}, \quad \kappa = \left(\frac{r_1}{r_2}\right)^{0.5}\sqrt{\frac{\omega_1\sigma_{NL}(\omega_2)}{\omega_2\sigma_{NL}(\omega_1)}} \tag{7d}$$

where $\widetilde{P}_{1,2}$ and $\widetilde{E}_0$ are the dimensionless slowly varying amplitudes of the particle dipole moments and external electric field, respectively, $\widetilde{\gamma}_{1,2}$ denotes the thermal/radiation losses of the nanosphere, $\widetilde{G}$ is the dimensionless coupling coefficient, and $\varsigma$ and $\kappa$ stand for the scale factors resulting from the asymmetry.

The analysis on these kinetic equations can firstly begin with the stationary states by considering $\mathrm{d}\widetilde{P}_{1,2}/\mathrm{d}\widetilde{t} = 0$ in Equation (6), i.e.,

$$\begin{cases} \left(i\widetilde{\gamma}_1 + W_1\widetilde{\Omega}_1 + |\bar{P}_1|^2\right)\bar{P}_1 + \widetilde{G}\bar{P}_2 = \widetilde{E}_0 \\ \left(i\widetilde{\gamma}_2 + W_2\widetilde{\Omega}_2 + \varsigma^2|\bar{P}_2|^2\right)\varsigma\bar{P}_2 + \kappa\widetilde{G}\bar{P}_1 = \kappa\widetilde{E}_0 \end{cases} \tag{8}$$

where $\bar{P}_{1,2}$ indicate the stationary solutions, and $\widetilde{E}_0$ only includes the background part $\widetilde{E}_b$. Here, we mainly focus on the regime of $W_{1,2}\widetilde{\Omega}_{1,2} < -\mathrm{Re}\widetilde{G} - \sqrt{3}\left|\widetilde{\gamma}_{1,2} - \mathrm{Im}\widetilde{G}\right|$ for the emergence of nontrivial solutions [21,46–48]. The further analysis on the linear stabilities of these stationary states involves the corresponding Jacobian matrix of Equation (6) when we separate the complex variables $\widetilde{P}_{1,2}$ into real and imaginary parts, which can be written as follows:

$$J = \begin{bmatrix} -W_1^{-1}A_1 & -W_1^{-1}O \\ -W_2^{-1}\kappa\varsigma^{-1}O & -W_2^{-1}A_2 \end{bmatrix} \tag{9}$$

Here,

$$A_n = \begin{bmatrix} 2\zeta_n^2 x_n y_n + \widetilde{\gamma}_n & W_n \widetilde{\Omega}_n + \zeta_n^2 (x_n^2 + 3y_n^2) \\ -\left[ W_n \widetilde{\Omega}_n + \zeta_n^2 (3x_n^2 + y_n^2) \right] & -2\zeta_n^2 x_n y_n + \widetilde{\gamma}_n \end{bmatrix} \tag{10}$$

$$O = \begin{bmatrix} \mathrm{Im}\widetilde{G} & \mathrm{Re}\widetilde{G} \\ -\mathrm{Re}\widetilde{G} & \mathrm{Im}\widetilde{G} \end{bmatrix} \tag{11}$$

where $x_n = \mathrm{Re}\widetilde{P}_n$, $y_n = \mathrm{Im}\widetilde{P}_n$, $n = 1, 2$, $\zeta_1 = 1$ and $\zeta_2 = \varsigma$. By calculating the eigenvalues of $J$ at the stationary point $(\overline{P}_1, \overline{P}_2)$, one can confirm that the stationary solution is stable only if the real parts of the eigenvalues are all negative. We further numerically develop Equation (6) with fourth-order Runge–Kutta scheme, and the corresponding temporal dynamical behaviors based on different parameters, initial conditions and external fields can be investigated in detail.

## 3. Results and Discussion

As for the universality of Equation (6) in representing a broad class of nonlinear coupled systems [18,19,39,49,50], we provide quantitative estimations of parameters without loss of generality in the basis of practicality. Here, we consider the geometric parameters of $r_1 = 100$ nm, $d = 300$ nm and variable $r_2$ which is the origin of asymmetric detuning in our model. The parameters of graphene are set as $\xi = 0.3$ ps, $E_F = 0.9$ eV and $v_F \approx c/300$, and the dielectric cores are set as $\varepsilon_1 = \varepsilon_2 = 2$. It should be remarked that the dielectric constant can correspond to $BaF_2$ whose dispersion is negligible in a terahertz range Ref. [51]. The host medium is set as $\varepsilon_h = 1$. According to Equation (4), we can obtain the resonant frequency $\hbar\omega_1 \approx 0.161$ eV of an isolated particle with $r_1 = 100$ nm. In order to simplify our discussion, the working frequencies of an external field illuminated on the asymmetric dimer are set with two discrete detuning values, i.e., $\widetilde{\Omega}_1 = (\omega - \omega_1)/\omega_1 = -0.1$ and $-0.04$. Furthermore, the evolution of forward scattering intensity is mainly concerned in our phase diagram for its macroscopic characteristics from the view of experimental accessibility. In the dipolar limit consideration, the expression of the dimensionless forward scattering intensity with $\widetilde{P}_{1,2}(\widetilde{t})$ parallel to the dimer axis and $kd \ll 1$ can be given by the following:

$$\widetilde{U}(\widetilde{t}) = \left| \widetilde{P}_1(\widetilde{t}) \right|^2 + \left| \widetilde{P}_2(\widetilde{t}) \right|^2 + 2 \left| \widetilde{P}_1(\widetilde{t}) \right| \left| \widetilde{P}_2(\widetilde{t}) \right| \cos\left[ \phi(\widetilde{t}) \right] \tag{12}$$

where $\phi(\widetilde{t})$ indicates the internal phase difference between the two dipoles.

In the first place, we investigate the situation with the strong detuning, i.e., $\widetilde{\Omega}_1 = -0.1$. As shown in Figure 2a, for the symmetric case, the linear instability analysis based on the Jacobian matrix $J$ in Equation (9) indicates that there is a lack of steady stationary solution in Regimes II and III for $\widetilde{E}_0$, roughly ranging from 0.0075 to 0.01073, and the bistability of $\widetilde{U}$ can only be detected in Regimes I and IV. For the asymmetric case shown in Figure 2b, the steady stationary solutions can be extended to all the regimes because of the inequality of detuning parameters, i.e., $\widetilde{\Omega}_1 \neq \widetilde{\Omega}_2$. In the meantime, by comparing Figure 2a,b, we can find that the bistable regimes are enlarged and accompanied with the emergence of tristability.

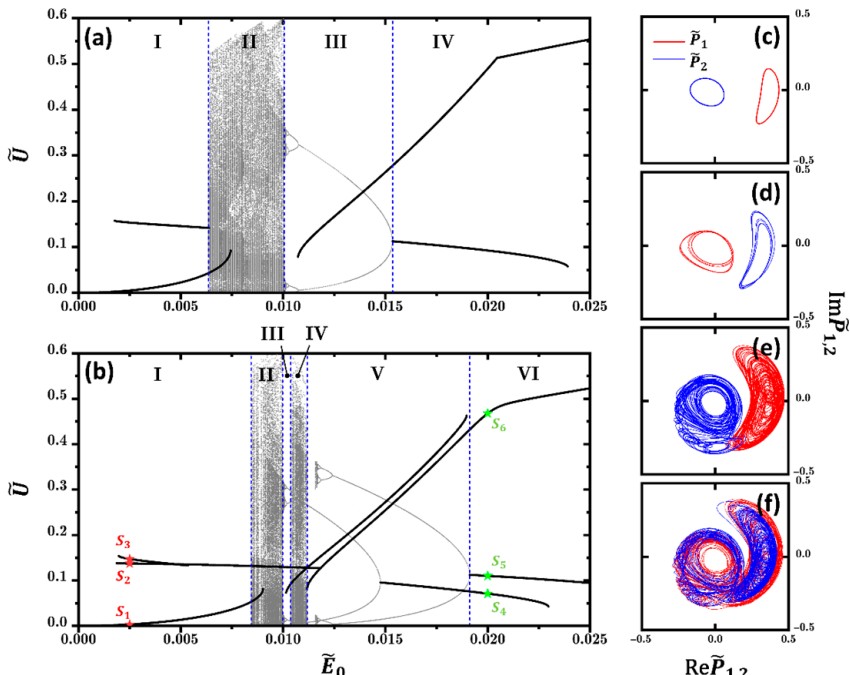

**Figure 2.** (**a**,**b**) Bifurcation diagrams based on $\widetilde{U}$ with strong detuning $\widetilde{\Omega}_1 = -0.1$, as well as (**a**) $r_2 = 100$ nm ($\widetilde{\Omega}_2 = -0.1$), and (**b**) $r_2 = 95$ nm ($\widetilde{\Omega}_2 = -0.1228$); the control parameter is the constant external electric field $\widetilde{E}_0$. The black lines indicate the stationary steady solutions and the gray dots indicate the unstable temporal solutions. With the vertical blue dashed lines, the diagrams are divided into several regimes, marked with Roman numerals. The color stars $S_{1-3}$ and $S_{4-6}$ represent the stationary solutions of $\widetilde{E}_0 = 0.0025$ and $\widetilde{E}_0 = 0.02$, respectively. (**c**–**f**) The typical phase portraits of temporal solutions extracted from Regimes V, III, IV and II shown in (**b**), which corresponds to (**c**) periodic self-oscillation state, (**d**) period doubling phenomenon and (**e**,**f**) chaos with $\widetilde{E}_0 =$ (**c**) 0.016, (**d**) 0.01008, (**e**) 0.01084 and (**f**) 0.0094, respectively.

Next, we investigate the characteristics of the temporal solutions, which result from the modulation instability. Here, we numerically calculate the temporal solutions of Equation (6) based on the fourth-order Runge–Kutta scheme. Simultaneously, we trace and denote the extrema of $\widetilde{U}(\tilde{t})$ using the Poincaré section method, and the results are presented as scatter gray points in Figure 2a,b. For the symmetric case, by decreasing $\widetilde{E}_0$, the lower branch of the bistable state undergoes an Andronov–Hopf bifurcation and generates a limit cycle or periodic self-oscillation state, which is depicted by two typical gray branches landing between Regimes III and IV ($\widetilde{E}_0 = 0.01535$), shown in Figure 2a. The further decrease in $\widetilde{E}_0$ in Figure 2a leads to an obvious period doubling on the self-oscillation depicted by the increase in the number of gray branches in Regime III, and the dimer system finally settles to a chaotic behavior which can manifest itself as non-periodic extrema of the temporal evolution in Regime II. However, the general scenario of temporal solutions based on asymmetric dimer can be more complicated. In Figure 2b, we can distinguish two Andronov–Hopf bifurcations with the decrease in $\widetilde{E}_0$; one begins at the boundary between Regimes V and VI ($\widetilde{E}_0 = 0.01912$), the other one begins in Regime V ($\widetilde{E}_0 = 0.01476$). Eventually, they both undergo period doubling bifurcation and reach the chaotic states shown in Regimes II and IV, respectively. Moreover, based on this symmetry breaking, it is found that one temporal solution can not only coexist with the steady stationary solutions, but also with other type of temporal solutions. For instance, the chaotic state, periodic self-oscillation state and two steady stationary solutions are coexistent in Regime IV, shown in Figure 2b.

To reveal the different dynamical behaviors in concrete details, we present the typical phase portraits extracted from the bifurcation diagram of the asymmetric case, which

correspond to the periodic self-oscillation (Figure 2c), period doubling phenomenon (Figure 2d), and chaotic behavior (Figure 2e,f), respectively. Here, we note that although the phase portrait of Figure 2e is not a typical strange attractor in butterfly-shape, its intrinsic dynamical behavior can be still chaotic. In order to describe these chaotic behaviors quantitatively, by applying the standard method [41], we have computed the corresponding four Lyapunov exponents $\Lambda_i(i = 1, 2, 3, 4)$, shown in Figure 2e, rising from the variable space $(x_1, y_1, x_2, y_2)$, and their values can be read as follows: 0.010742, $-0.000031$, $-0.014132$, and $-0.024261$, respectively. With these values, we can obtain the Kaplan–Yorke (KY) fractal dimension $D_{KY} = 2.441493$ using the method described in Ref. [52], which is an obvious signature of the chaotic behavior when $D_{KY} > 2$. The formula of $D_{KY}$ is as follows:

$$D_{KY} = j + \frac{1}{\left|\Lambda_{j+1}\right|} \sum_{i=1}^{j} \Lambda_i \tag{13}$$

where $j$ is the largest integer satisfying $\sum_{i=1}^{j} \Lambda_i \geq 0$ and $\sum_{i=1}^{j+1} \Lambda_i \leq 0$, with $\Lambda_1 > \Lambda_2 > \Lambda_3 > \Lambda_4$. Similarly, for the chaotic behavior shown in Figure 2f, we expectedly obtain $D_{KY} = 2.479822 > 2$. In our following results and discussions, the chaotic behaviors are all confirmed with this procedure.

When we consider the relatively small detuning parameter, i.e., $\widetilde{\Omega}_1 = -0.04$, it can be found that the temporal solution and the steady stationary state can be completely separated in the bifurcation diagram in the symmetric case (Figure 3a), and the bistability of $\widetilde{U}$ can only be detected in Regime V, in Figure 3a. When $\widetilde{\Omega}_1 \neq \widetilde{\Omega}_2$, similar with Figure 2b, the asymmetry can induce the extension of the regimes of steady stationary solutions, as well as the emergence of bistability with a low switch-threshold electric field and multistability shown in Regimes I and V in Figure 3b, respectively.

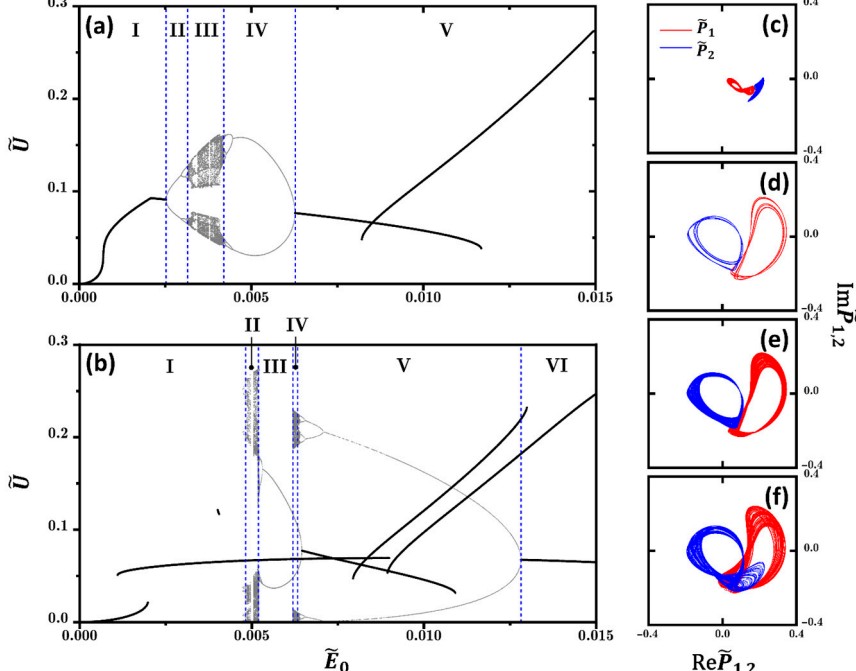

**Figure 3.** (**a**,**b**) Bifurcation diagrams based on $\widetilde{U}$ with relatively small detuning $\widetilde{\Omega}_1 = -0.04$, as well as (**a**) $r_2 = 100$ nm ($\widetilde{\Omega}_2 = -0.04$), and (**b**) $r_2 = 90$ nm ($\widetilde{\Omega}_2 = -0.0893$); the control parameter is the constant external electric field $\widetilde{E}_0$. The black lines indicate the stationary steady solutions and the gray dots indicate the unstable temporal solutions. With the vertical blue dashed lines, the diagrams are divided into several regimes, marked with Roman numerals. (**c**–**f**) The phase portraits of temporal solutions extracted from Regimes III of (**a**), V, IV and II of (**b**), which correspond to (**c**,**e**,**f**) chaos and (**d**) period doubling phenomenon with $\widetilde{E}_0 = $ (**c**) 0.0036, (**d**) 0.0064, (**e**) 0.0062 and (**f**) 0.0051, respectively.

The further illustration on the temporal solutions has followed the same trend in Figure 2. Here, we notice that the chaotic regime is between the two regimes of periodic self-oscillation for the symmetric case shown in Figure 3a. This means that the chaotic behavior will not disappear abruptly, i.e., undergo other types of temporal solutions before turning into the steady stationary solutions as the decrease in $\widetilde{E}_0$, which is quite different from the phenomenon at the boundary between Regimes I and II of Figure 2a,b and Figure 3b. When asymmetry is introduced in this dimer system, similarly, we can observe that the number of Andronov–Hopf bifurcation is doubled: one is at $\widetilde{E}_0 = 0.01282$, the other is at $\widetilde{E}_0 = 0.00646$. Also, as the decrease in $\widetilde{E}_0$, they both lead to the emergence of the chaotic states shown in Regimes II and IV in Figure 3b, respectively, whose regime is much narrower than the one in asymmetric case with strong detuning. Here, we also plot the phase portraits extracted from the typical regimes. We can confirm that Regime III of the symmetric case (Figure 3c), and Regimes II and IV of the asymmetric case (Figure 3e,f) are all chaotic behaviors according to the calculation on $D_{KY}$, while the phase portrait in Figure 3d is the period doubling phenomenon. In addition, this symmetry breaking with the inequality of detuning parameters can also result in the coexistence of temporal solutions and steady stationary solutions under the same external electric field.

To further reveal the coexistence of temporal dynamic and stationary steady solutions, as well as the switching among them, we introduce a hard excitation whose envelope is in a form of quasi-square function, and its formula can be expressed as follows:

$$\widetilde{E}_{pulse}\left(\widetilde{t}\right) = \frac{E_p}{\pi}\left[\mathrm{atan}\frac{\widetilde{t}-\widetilde{t}_0}{\widetilde{\tau}} - \mathrm{atan}\frac{\widetilde{t}-\widetilde{t}_0-\Delta\widetilde{t}}{\widetilde{\tau}}\right] \qquad (14)$$

where $E_p$, $\widetilde{t}_0$, $\Delta\widetilde{t}$ and $\widetilde{\tau}$ represent the saturation value, starting time, duration and edge sharpness of the hard excitation pulse, respectively. Hence, the total external field during our following calculation should be a combination of the background field $\widetilde{E}_b$ and the pulse signal $\widetilde{E}_{pulse}$, which results in the time-dependent $\widetilde{E}_0$, i.e., $\widetilde{E}_0\left(\widetilde{t}\right) = \widetilde{E}_b + \widetilde{E}_{pulse}\left(\widetilde{t}\right)$.

For the switching among the stationary steady solutions, we consider the cases of tristability extracted from Regimes I and VI in Figure 2b, and the background fields $\widetilde{E}_b$ are set as 0.0025 and 0.02, corresponding to the red and green star points shown in Figure 2b, respectively. By initializing the numerical calculation with the lower state $S_1$, we can find the scenario of indefinite switching among the stationary steady solutions $S_{1-3}$, depicted in Figure 4a with $\widetilde{E}_b = 0.0025$. The result means that although the total external field intensity $\widetilde{E}_0$ is considerably larger than the corresponding upper thresholds of the tristability, the dimer system can eventually either transit to one of the upper states $S_{2,3}$ or remain in the initial state $S_1$, sensitively depending both on $E_p$ and $\Delta\widetilde{t}$. In Figure 4b, for the case of $\widetilde{E}_b = 0.02$, a similar scenario is detected when the initial lower state is set as $S_4$, but the domain of final upper state $S_6$ is quite dominant because of the much higher $\widetilde{E}_b$. Examples of such indefinite switching are shown in Figure 4c–f. By comparison, we can find that the switching to upper states can be cancelled (as the blue lines shown in Figure 4d,f), even though the corresponding duration $\Delta\widetilde{t}$ or saturation value $E_p$ is not the smallest one.

Remarkably, the situation of switching among the temporal dynamic solutions and stationary steady solutions can be more interesting and complicated because of their coexistence in the same regime. As shown in Figure 5, for the strong detuning and asymmetric case with $\widetilde{E}_b = 0.013$ (from Regime V in Figure 2b), when we initialize the dimer system with a periodic self-oscillation solution (Figure 5a,b), it can switch into a stationary steady state or another periodic self-oscillation solution with larger amplitude depending on the duration $\Delta\widetilde{t}$ of the hard excitation despite the same $E_p$. In the meantime, the initialized stationary steady state can also switch into the periodic self-oscillation solutions with different amplitudes, respectively, by applying the signals with the same $\Delta\widetilde{t}$ but different $E_p$. When the system involves the chaotic solution with $\widetilde{E}_b = 0.011$, (from Regime IV in Figure 2b), the similar indefinite switching among the temporal dynamic solutions and stationary steady solutions is also detected in Figure 6, which is sensitively determined by the profile of the

hard excitation. In addition to the generation of self-oscillations or chaos from the same stationary steady solution shown in Figures 6a and 6b, respectively, one should especially notice the switching between self-oscillation and chaos shown in Figure 6c,d. Practically, it provides an efficient route to exchange the radiation system from an oscillator to a random generator and vice versa, which is unachievable for the symmetrical dimer because of the lack of coexistence of temporal dynamic solutions [18,19,49,50].

In addition, we should mention that via an in-depth investigation of the different coexistent regimes of this asymmetric dimer system, one can thoroughly establish the mapping relationship between the profiles of external stimulation and scattering response, which may provide potential applications based on the tunable and exchangeable modulation instability.

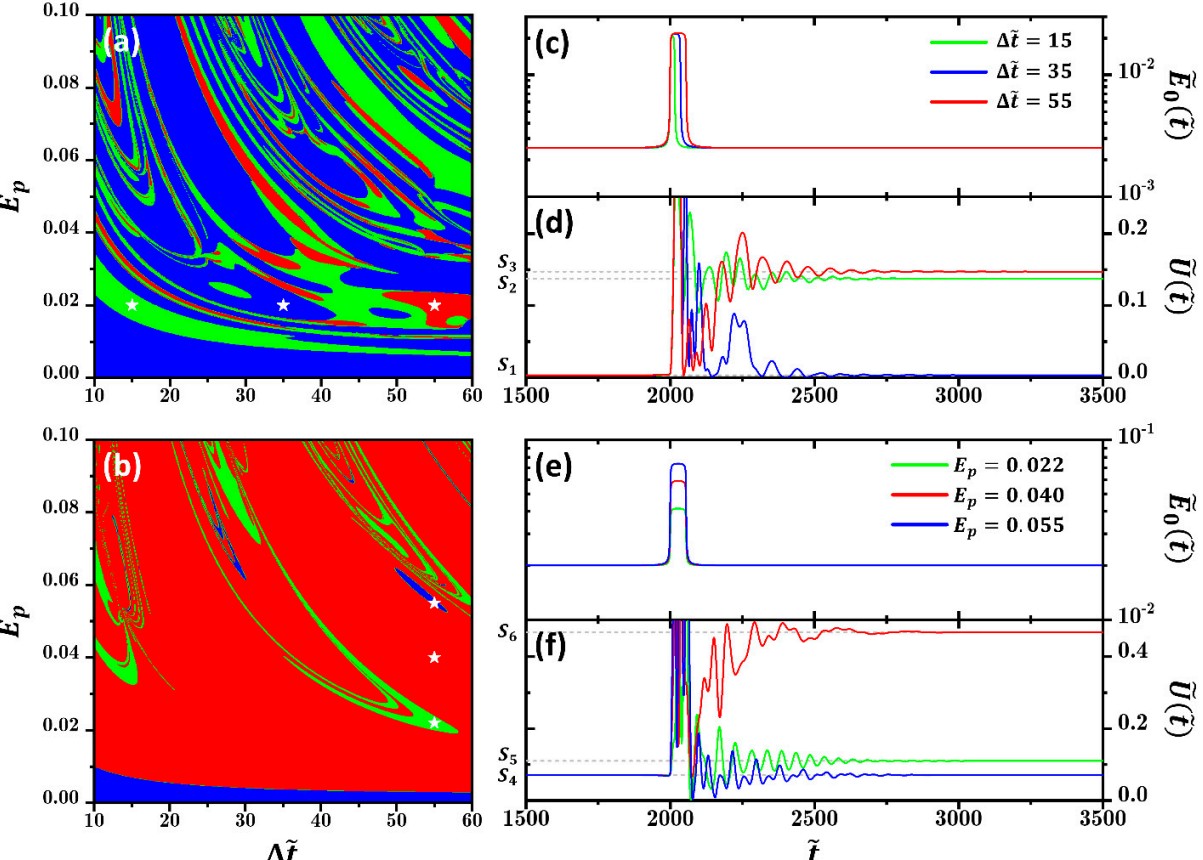

**Figure 4.** (**a**,**b**) The switching diagrams in terms of pulse saturation value $E_p$ and duration $\Delta\tilde{t}$ based on background field (**a**) $\tilde{E}_b = 0.0025$ and (**b**) $\tilde{E}_b = 0.02$, respectively; the colors indicate different final stationary states, respectively, with blue ($S_{1,4}$), green ($S_{2,5}$) and red ($S_{3,6}$). The white star points correspond to the parameters of the examples shown in (**c**,**d**) and (**e**,**f**). (**c**) The input signal $\tilde{E}_0(t)$ with varied $\Delta\tilde{t}$ and fixed $E_p = 0.02$. (**d**) The temporal evolutions correspond to the input signal of (**c**). The gray dashed lines indicate the scattering intensities of $S_{1-3}$. (**e**) The input signal $\tilde{E}_0(t)$ with varied $E_p$ and fixed $\Delta\tilde{t} = 55$. (**f**) The temporal evolutions correspond to the input signal of (**e**). The gray dashed lines indicate the scattering intensities of $S_{4-6}$.

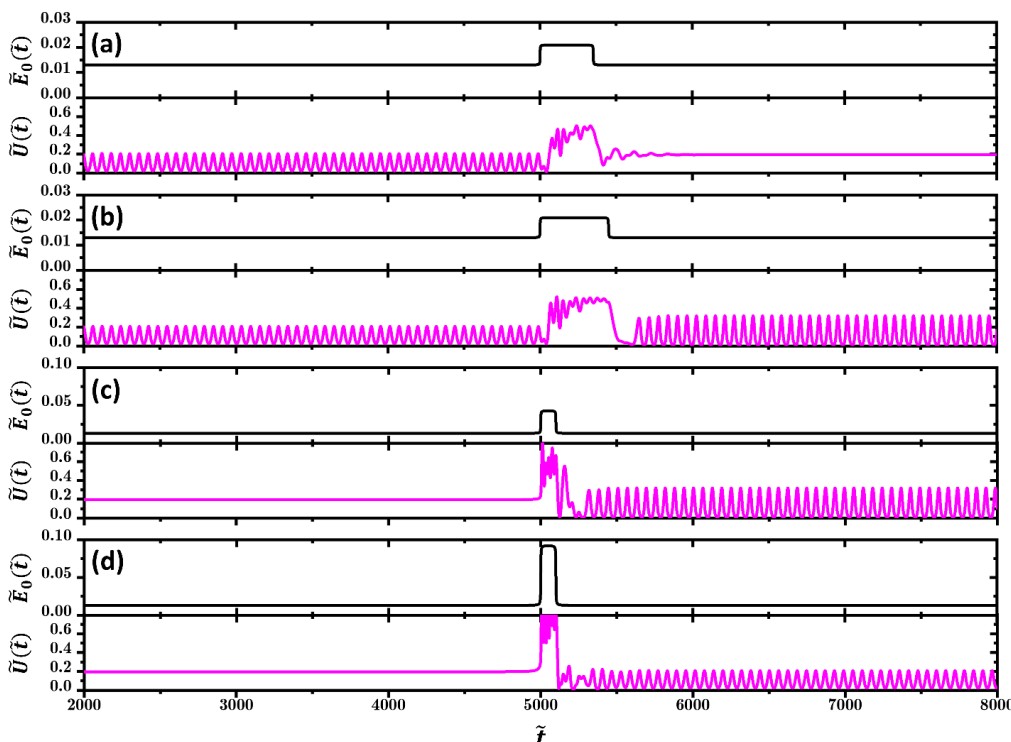

**Figure 5.** The black lines indicate the total external field $\widetilde{E}_0(\widetilde{t})$, with $\widetilde{E}_b = 0.013$ and (**a**) $E_p = 0.008$, $\Delta\widetilde{t} = 350$; (**b**) $E_p = 0.008$, $\Delta\widetilde{t} = 450$; (**c**) $E_p = 0.03$, $\Delta\widetilde{t} = 100$; and (**d**) $E_p = 0.08$, $\Delta\widetilde{t} = 100$. The pink lines represent the temporal evolutions of $\widetilde{U}(\widetilde{t})$ corresponding to the input signals shown in (**a**–**d**), respectively. The parameters of the dimer system are $\widetilde{\Omega}_1 = -0.1$ and $r_2 = 95$ nm ($\widetilde{\Omega}_2 = -0.1228$).

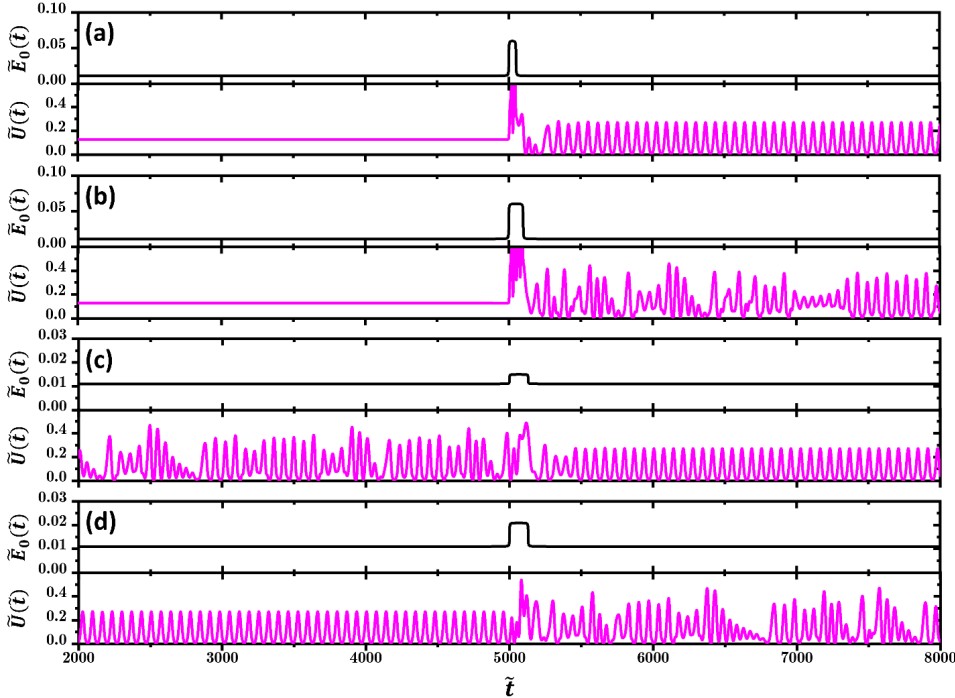

**Figure 6.** The black lines indicate the total external field $\widetilde{E}_0(\widetilde{t})$, with $\widetilde{E}_b = 0.011$ and (**a**) $E_p = 0.05$, $\Delta\widetilde{t} = 50$; (**b**) $E_p = 0.05$, $\Delta\widetilde{t} = 100$; (**c**) $E_p = 0.004$, $\Delta\widetilde{t} = 130$; and (**d**) $E_p = 0.01$, $\Delta\widetilde{t} = 130$. The pink lines represent the temporal evolutions of $\widetilde{U}(\widetilde{t})$ corresponding to the input signals shown in (**a**–**d**), respectively. The parameters of the dimer system are $\widetilde{\Omega}_1 = -0.1$ and $r_2 = 95$ nm ($\widetilde{\Omega}_2 = -0.1228$).

## 4. Conclusions

To summarize, we have proposed a nonlinear nanodimer system made of a pair of graphene-wrapped dielectric nanospheres, whose resonant frequency is tunable by changing the radius of the nanosphere. The detailed analytical and numerical investigations on the bifurcation diagram indicate that the dimer system can present different temporal or stationary solutions, such as multistability, periodic self-oscillation, period doubling phenomenon and chaos. These solutions can be spontaneously stimulated by applying the external field only with the background part. We find that the introduction of asymmetry into the dimer system can increase the number of regimes with temporal solutions, as well as result in the emergence of regimes with multistability and the expansion of regimes with stationary steady solutions. By calculating the Lyapunov exponent and fractal dimension [41,52], we can quantitatively identify the main characteristics of chaotic behaviors from the temporal solutions in the bifurcation diagram. It is also presented that the phase portraits of chaos can be quite different in topology, although they can always provide $D_{KY} > 2$. Furthermore, with a hard excitation in quasi-square form combined with the background field, we demonstrate that the indefinite switching is a universal phenomenon. This kind of switching can occur not only among the stationary steady solutions but also among the stationary and temporal solutions because of the coexistence of these solutions. We also find that the switching can be sensitively dependent on the saturation value and the duration of the hard excitation. In terms of practical accessibility, the evolution of the scattering intensity of the dimer system is mainly concerned, and the efficient exchange among the self-oscillation and the chaos can provide possible applications as a nanoantenna. We also hope that our findings on this temporal dynamical dimer system with nonlinearity can provide insight into the design of on-chip nanophotonic devices with tunable functionalities, such as logical operator, astable multivibrator, random number generator, and so on.

**Author Contributions:** Conceptualization, X.J. and L.G.; methodology, X.J.; formal analysis, X.J.; investigation, X.J.; writing—original draft preparation, X.J.; writing—review and editing, X.J., Y.H., P.M., A.S.S. and L.G. All authors have read and agreed to the published version of the manuscript.

**Funding:** The authors gratefully acknowledge the financial support from the National Natural Science Foundation of China (Grant Nos. 92050104, 12274314, 12174281), and the Natural Science Foundation of the Jiangsu Province (Grant No. BK20221240). A. S. Shalin gratefully acknowledges the financial support from the Ministry of Science and Higher Education of the Russian Federation (Agreement No. 075-15-2022-1150) and the support of the Latvian Council of Science (project: DNSSN, No. lzp-2021/1-0048). The investigation of time-dependent response has been partially supported by the Russian Science Foundation (Grant No. 21-12-00151).

**Institutional Review Board Statement:** Not applicable.

**Informed Consent Statement:** Not applicable.

**Data Availability Statement:** The data that support the findings of this study are available from the corresponding author upon reasonable request.

**Conflicts of Interest:** The authors declare no conflict of interest.

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
