# Peer review of "Temporal Dynamics of an Asymmetrical Dielectric Nanodimer Wrapped with Graphene"

_photonics, doi:10.3390/photonics10080914_

Round 1

Reviewer 1 Report

In the article entitled "The Temporal Dynamics of Asymmetrical Dielectric Nanodimer Wrapped with Graphene", the authors developed a nonlinear dynamical model of asymmetrical nanodimer with the dispersion relation method. In the bifurcation diagram, they presented that the scattering intensity of the system can spontaneously be either stationary steady or temporal dynamical at the same background external field. The introduced symmetry-breaking can lead to the increase of number of regimes with temporal solutions (periodic self-oscillation, period doubling phenomenon and chaos), as well as the emergence of the regimes with multistable stationary steady solutions. This work also indicates that the indefinite switching can be generally triggered by applying different pulse signals not only among the stationary steady solutions, but also among all the possible solutions. The methods employed and the results obtained are consistent, and they may provide a potential insight in the design of nonlinear nanoantenna. I consider this submission interesting and worthy of publication in Photonics with some minor revisions: 

1)      As the quantitative mathematical tool from nonlinear dynamical theory, a brief but detail description on the definition of Kaplan–Yorke fractal dimension should be introduced in consideration of easy reading.

2)      Although the graphene as a material has implicitly included the linear and nonlinear part, the author may include nonlinearity as a keyword in the paper for the further precise searching and citation.

3)      In fact, the scheme proposed by the authors as the nanoantenna can be a topic with widespread interest. I suggest that the authors added more closely related recent works of nanoantenna and particle scattering.

4)      The permittivity of BaF2 is considered in the terahertz frequency regime, but the corresponding reference for this parameter is not included in the manuscript.

5)      For all the formulas, please confirm that all the dimensionless variables are denoted in a same style.

The English language quality of the current version meets the publication requirement.

Reviewer 2 Report

Comments are in attached .pdf file. 

Reviewer 3 Report

The present manuscript is a theoretical and numerical work to study the temporal dynamics of a nonlinear nanodimer system, which is made of a pair of graphene-wrapped dielectric nanospheres. The authors derive the temporal kinetic equations in asymmetric form with symmetry breaking on the resonant frequencies. The bifurcation diagrams and dynamical behaviors are shown by the instability analysis and numerical results. The authors also investigate the characteristics of indefinite switching, including stationary steady solutions and the temporal solutions based on the coexistence.

This work will arise the interest of nonlinear nanophotonics, particular for the further step to some promising applications of on-chip tunable functionalities. The results are solid and convinced. I think the present manuscript has reached the quality of Photonics, so I would be happy to recommend publication in this version.

Author Response

We thank the reviewer for endorsing our work. And We are glad to have the present manuscript reaching the quality of Photonics.

Reviewer 4 Report

Dear Editor and Authors:

I have carefully read the manuscript titled “The Temporal Dynamics of Asymmetrical Dielectric Nanodimer Wrapped with Graphene” by Jiang and colleagues. The authors describe a nanophotonic device that can scatter light in the THz region with distinctive temporal dynamics, even when subjected to continuous wave excitation. In particular, the authors have designed a dimer of dielectric nanoparticles wrapped with a layer of graphene. Thanks to the Kerr nonlinearity of graphene, this device can undergo a phase transition to a self-oscillating state. The authors thoroughly evaluate the different dynamics and stationary states that can be obtained by using well-known techniques on dynamical systems, such as bifurcation maps, Lyapunov exponents, etc… By introducing an asymmetry in the sizes of the nanoparticles in the dimer, the authors manage to modify the system so that different temporal behaviours coexist for the same excitation field strength. Finally, the authors clearly demonstrate that these different regimes can be accessed using the well-known hard-excitation approach.

Designing a photonic system that can deliver different dynamics in a controllable fashion is always good news, as these devices will have far-reaching implications in photonic circuits for all-optical information processing. In that regard, the results reported in this paper are interesting and suggest a smart approach. The methodology is mostly understandable (but for a few aspects, see below), and the results seem technically sound and correct. The paper is mostly well-written (but for a few typos, see below), well organized, and can be understood by researchers working in the field of photonics and dynamical systems. Accordingly, I could recommend publication of this manuscript in the journal MDPI Photonics.

Comments and suggestions:

-Could the authors please specifically write down the expression for the polarizabilities of the individual graphene-wrapped nanoparticles alpha_1,2 and how to derive it (adding a reference is ok)? That would help a lot in following the derivation, specially to check eq. (3).

-Right before eq. (8), the manuscript states “The further analysis on the linear stabilities of these stationary states involves the corresponding Jacobian matrix of Eq. (5) when we separate the complex variables \tilde{P}_{1,2} into real and imagine parts”. Attending to the expressions that follows this equation, are the authors representing the Jacobian matrix of Eq. (5) or Eq. (7)? If that is the case, it would be \bar{P}_{12} and not \tilde{P}_{1,2}, wouldn’t it?

-In that same sentence, “imagine parts” should be “imaginary parts”.

-The sentence right after Eq. (12) shouldn’t be there. Please, remove.
